# Dual Action of the PN159/KLAL/MAP Peptide: Increase of Drug Penetration across Caco-2 Intestinal Barrier Model by Modulation of Tight Junctions and Plasma Membrane Permeability

**DOI:** 10.3390/pharmaceutics11020073

**Published:** 2019-02-10

**Authors:** Alexandra Bocsik, Ilona Gróf, Lóránd Kiss, Ferenc Ötvös, Ottó Zsíros, Lejla Daruka, Lívia Fülöp, Monika Vastag, Ágnes Kittel, Norbert Imre, Tamás A. Martinek, Csaba Pál, Piroska Szabó-Révész, Mária A. Deli

**Affiliations:** 1Institute of Biophysics, Biological Research Centre, Hungarian Academy of Sciences, H-6726 Szeged, Hungary; bocsik.alexandra@brc.mta.hu (A.B.); ilona.grof@brc.mta.hu (I.G.); kiss.lorand@med.u-szeged.hu (L.K.); 2Doctoral School in Biology, Faculty of Science and Informatics, University of Szeged, H-6726 Szeged, Hungary; daruka.lejla@brc.mta.hu; 3Department of Pathophysiology, University of Szeged, H-6701 Szeged, Hungary; 4Institute of Biochemistry, Biological Research Centre, Hungarian Academy of Sciences, H-6726 Szeged, Hungary; otvos.ferenc@brc.mta.hu; 5Plant Biology, Biological Research Centre, Hungarian Academy of Sciences, H-6726 Szeged, Hungary; zsiros.otto@brc.mta.hu; 6Synthetic and Systems Biology Unit, Institute of Biochemistry, Biological Research Centre, Hungarian Academy of Sciences, H-6726 Szeged, Hungary; pal.csaba@brc.mta.hu; 7Department of Medical Chemistry, University of Szeged, H-6720 Szeged, Hungary; fulop.livia@med.u-szeged.hu (L.F.); imre.norbert@med.u-szeged.hu (N.I.); martinek.tamas@med.u-szeged.hu (T.A.M.); 8Division of Pharmacology and Drug Safety Research, Gedeon Richter Plc., H-1103 Budapest, Hungary; m.vastag@richter.hu; 9Institute of Experimental Medicine, Hungarian Academy of Sciences, H-1450 Budapest, Hungary; kittel@koki.hu; 10Department of Pharmaceutical Technology, University of Szeged, H-6720 Szeged, Hungary; revesz@pharm.u-szeged.hu

**Keywords:** absorption enhancer, antimicrobial peptide, Caco-2, claudin, cell-penetrating peptide (CPP), drug delivery, intestinal epithelial cells, KLAL, PN159, tight junction modulator

## Abstract

The absorption of drugs is limited by the epithelial barriers of the gastrointestinal tract. One of the strategies to improve drug delivery is the modulation of barrier function by the targeted opening of epithelial tight junctions. In our previous study the 18-mer amphiphilic PN159 peptide was found to be an effective tight junction modulator on intestinal epithelial and blood–brain barrier models. PN159, also known as KLAL or MAP, was described to interact with biological membranes as a cell-penetrating peptide. In the present work we demonstrated that the PN159 peptide as a penetration enhancer has a dual action on intestinal epithelial cells. The peptide safely and reversibly enhanced the permeability of Caco-2 monolayers by opening the intercellular junctions. The penetration of dextran molecules with different size and four efflux pump substrate drugs was increased several folds. We identified claudin-4 and -7 junctional proteins by docking studies as potential binding partners and targets of PN159 in the opening of the paracellular pathway. In addition to the tight junction modulator action, the peptide showed cell membrane permeabilizing and antimicrobial effects. This dual action is not general for cell-penetrating peptides (CPPs), since the other three CPPs tested did not show barrier opening effects.

## 1. Introduction

The oral administration of medicines is the most common method in drug therapy because of its easy administration and good patient compliance. However, the absorption of several drugs is limited in the gastrointestinal tract by the barriers composed of epithelial cell layers [1]. Since the non-invasive delivery of orally administered hydrophilic drugs or biopharmaceuticals to the systemic circulation is still a challenge, several strategies and molecules have been investigated as penetration enhancers, including biosurfactants, sucrose esters and peptides [1,2,3].

Tight junctions (TJs) between epithelial cells form the anatomical basis of the intestinal barrier which is one of the main biological barriers in our organism and the body’s largest interface with the external environment [1]. These TJs, located in the apical regions of cells, close the intercellular gap and thus determine the paracellular permeability and the tightness of the epithelial barriers [1,4]. The absorption level and rate across biological barriers depends on the composition of cellular membranes, the anatomical structures, and the expression pattern of TJ proteins and transporters in the cell membrane [5]. TJs are complex structures composed of integral membrane proteins including among others the occludin protein and the claudin family [6].

One of the basic strategies to improve drug delivery is the modulation of barrier function by the targeted opening of TJs [1] Peptides which act directly on TJs and modulate their permeability are potential candidates to increase the absorption of hydrophilic or large therapeutic molecules in a non-invasive manner [1,7]. In our recent work six TJ modulator peptides acting on different targets were compared using culture models of the intestinal and the blood-brain barriers [8]. All the six peptides induced reversible opening of TJs but barrier selectivity and differences in efficacy were observed. We found the PN159 as the most effective TJ modulator peptide on both barrier models [8]. This peptide was identified as a TJ modulator on cultured human bronchial epithelial cell layers in 2005 [4]. PN159 peptide effectively increased the permeability of respiratory epithelium both in culture and animal experiments [9]. The specific target proteins of the peptide were unknown, but we demonstrated that PN159 interacts with the extracellular loops of integral membrane TJ proteins claudin-1 and -5 [8].

The amino acid sequence of the 18-mer PN159, KLALKLALKALKAALKLA-amide, contains several KLA motifs, and therefore it is also named as KLA or KLAL peptide [10]. The other name of this peptide is MAP, which stands for model amphipathic peptide [11,12]. The structure of the lipid vesicle-bound PN159 peptide is mostly α-helical [10] and strongly interacts with biological membranes [10,13]. Cell-penetrating peptides (CPPs) are a group of short natural or synthetic peptides with 5–30 amino acids, which by crossing cellular membranes can transport into cells a wide range of small and macromolecules, including nucleic acids, proteins, imaging agents [14]. Based on their physical-chemical properties, CPPs can be classified into groups. The most investigated CPPs belong to the amphipathic class. These peptides contain hydrophilic and hydrophobic motifs and are especially rich in lysine, arginine, leucine and alanine. Examples include the PN159 peptide [10,11]; the cyclic φ-peptide [15] and Pep-1 [16]. Among the cationic CPPs penetratin [17] or polyarginine peptides [18] are extensively investigated.

CPPs are structurally similar to antimicrobial peptides, which are important defense elements of the innate immunity, and accordingly, they show an antimicrobial effect against pathogens [19]. Bacterial resistance is a growing clinical problem, as it was highlighted by the Infectious Diseases Society of America publishing the list of ESKAPE pathogens (*Enterobacter species, Staphylococcus aureus, Klebsiella pneumoniae, Acinetobacter baumannii, Pseudomonas aeruginosa* and *Enterococcus faecium*) [20]. This group of microbes causes the large majority of nosocomial infections throughout the world. The combat antibiotic resistance, CPPs can be promising alternative molecules, since increased sensitivity (collateral sensitivity) for antimicrobial peptides were found in antibiotic-resistant *Escherichia coli* strains [21].

As a culture model of the intestinal epithelial barrier, we used in our study the Caco-2 human cell line resembling the epithelium of the small intestine both from structural and functional aspects [22]. The cells have polarized cell morphology, grow in monolayer, possess microvilli, form TJs, express nutrient and efflux transporters, and show good correlation with in vivo data [23,24]. Caco-2 epithelial cells are routinely used in drug permeability studies [24,25].

Crucial parameters for absorption enhancers include their safety, reversibility and efficacy. There are no data available about the effectiveness and safety of PN159 peptide on the intestinal barrier, so our primary goal was to test the TJ modulator peptide for these aspects. Therefore, the aim of the study was to (i) determine the influence of long-time and concentration-dependent effects of treatments with PN159 peptide on intestinal epithelial cell viability, barrier function and recovery; (ii) test the effect of PN159 peptide on drug penetration across the intestinal barrier model; (iii) identify further potential targets of this TJ modulator peptide by molecular modelling; (iv) measure the cell uptake of the PN159 in intestinal epithelial cells and its antimicrobial activity on ESKAPE pathogens; and (iv) test other CPPs for the TJ modulator effect.

## 2. Materials and Methods

### 2.1. Materials

All reagents were purchased from Sigma-Aldrich Ltd. (Budapest, Hungary) except for those specifically mentioned.

### 2.2. Peptide Synthesis

PN159 peptide (KLALKLALKALKAALKLA-amide) [4,10], and Pep-1 (Chariot) peptide (KETWWETWWTEWSQPKKKRKV-amide) were synthesized manually on a 0.5 mmolar scale with the use of standard Fmoc-chemistry on a Rink-amide resin. Couplings were performed in DMF with three-fold excess of DCC, HOBt, and Fmoc-amino acids for 3 h at ambient temperature. In the case of octaarginine (RRRRRRRR-amide, R8) three-fold excess of HATU and six-fold excess of DIPEA was used. Fmoc deprotection was performed in 20% piperidine/DMF mixture for 20 min. The peptides were cleaved from the resin by incubating them with the mixture of TFA/water/triisopropylsilane (48:1:1 volume ratio), precipitated with diethyl-ether and lyophilized. Crude peptides were purified using a Shimadzu semi-preparative high-performance liquid chromatography (HPLC) instrument equipped with a Phenomenex JupiterC18 column, in the following solvent system: (A) 0.1% aqueous TFA and (B) 0.1% TFA in 80% aqueous acetonitrile, in a linear gradient mode. Analysis and purity control were carried out on an analytical HPLC instrument (HP Model 1100 liquid chromatograph equipped with a Phenomenex Jupiter C18 column). Quality control of the peptides was done by performing mass spectrometric measurements on a FinniganTSQ-7000 triple quadrupole mass spectrometer in positive ion mode.

The cyclic φ-peptide (cyclo[CGGFWRRRRGE(εAca)G])was also synthesized manually on a 0.5 mmolar scale with the use of Boc-chemistry on a MBHA-HCl resin, by applying a native chemical ligation strategy. Couplings were performed in DMF with three-fold excess of DIC, HOBt, and Boc-amino acids for 3 h at ambient temperature. Boc deprotection was performed in TFA/DCM (1:1 volume ratio) mixture for 20 min. The peptide was cleaved from the resin by the standard HF method. Native chemical ligation was performed with 2% thiophenol in an ammoniumacetate solution (0.1 M) at room temperature for 12h. Cyclic crude peptide was purified and analyzed as described above.

### 2.3. Cell Culture

The human Caco-2 intestinal epithelial cell line was purchased from ATCC (cat.no. HTB-37). Caco-2 cells were grown in DMEM/HAM’s F-12 culture medium with stable glutamine (Life Technologies, Gibco, Carlsbad, CA, USA) supplemented with 10% fetal bovine serum (Life Technologies, Gibco, Carlsbad, CA, USA and 50 μg/mL gentamycin in a humidified incubator with 5% CO_2_ at 37 °C. All plastic surfaces were coated with 0.05% rat tail collagen in sterile distilled water before cell seeding.

### 2.4. Peptide Treatment

The PN159 peptide stock solution (5 mM) was prepared freshly in sterile DMSO. Treatment solutions were further diluted in Ringer-Hepes (150 mM NaCl, 6 mM NaHCO_3_, 5.2 mM KCl, 2.2 mM CaCl_2_, 0.2 mM MgCl_2_, 2.8 mM d-glucose, 5 mM Hepes; pH 7.4) or cell culture medium. Final concentrations of the peptide in treatment solutions were as follows: 1, 3, 10, 30 and 100 µM for cell viability assays; 1, 3, 10 and 30 µM for the barrier integrity assays and 10 µM for the recovery measurements.

The stock solutions of the three reference CPPs were the following: Pep-1: 1 mM in distilled water (DW); R8: 1 mM in DW; φ-peptide, 5 mM in DW and 2.5% DMSO. The 100 µM working solutions were dissolved in cell culture medium or Ringer–Hepes buffer. We prepared all solutions freshly in sterile conditions.

### 2.5. MTT Assay

For the cell viability assays Caco-2 epithelial cells were seeded to 96-well plates at the density of 6 × 10^3^ cells/well (Orange Scientific) and cultured for 3-5 days. Confluent cultures were treated with PN159 peptide (1, 3, 10, 30 and 100 μM) in phenol red free DMEM (Life Technologies, Gibco, Carlsbad, CA, USA) for 1-h. This viability assay was performed at three different time points: (i) immediately after the 1-h treatment, (ii) at one-day recovery, (iii) at one-week recovery. To determine 100% toxicity, cells were incubated with 1 mg/mL Triton X-100 detergent. After treatment the medium was changed and 0.5 mg/mL MTT (3-(4,5-dimethyltiazol-2-yl)-2,5-diphenyltetrazolium bromide) solution was added to the cells for 3 h. During the incubation cells were kept in a CO_2_ incubator. The metabolic activity of cells is reflected by the conversion of the yellow MTT dye to purple formazan. Formazan crystals were dissolved in DMSO and the amount of converted dye was determined by measuring absorbance at 595 nm wavelength with a microplate reader (Fluostar Optima, BMG Labtechnologies, Ortenberg, Germany). Cell viability and/or metabolic activity was calculated as percentage of dye conversion by non-treated cells.

### 2.6. Impedance Measurement

Kinetics of epithelial cell reaction to PN159 peptide treatment was monitored by impedance measurement at 10 kHz (RTCA-SP instrument, ACEA Biosciences, San Diego, CA, USA). Impedance measurement is non-invasive, label-free and real time, and linearly correlates with adherence, growth, number and viability of cells [2,26]. For background measurements 50 μL cell culture medium was added to the wells, then cells were seeded at a density of 6 × 10^3^ cells/well to the coated 96-well plates with integrated gold electrodes (E-plate 96, ACEA Biosciences). Cells were cultured for 5–7 days in a CO_2_ incubator at 37 °C and monitored every 10 min until the end of experiments. Cells were treated with PN159 peptide solutions at 1, 3, 10, 30 and 100 μM concentrations at the beginning of the plateau phase of growth and the effects were followed for 1-h or 24 h. To measure cell recovery, peptide solutions were changed to culture medium and the impedance was monitored for one week. Culture medium was changed every 2 days. Caco-2 cells were also treated with the three selected CPPs, Pep-1, R8 and φ-peptide, at 100 μM concentrations at the beginning of the plateau phase of cell growth for 24-h. Triton X-100 detergent (1 mg/mL) was used as a reference compound to induce cell death. Cell index was defined as *R*_n_-*R*_b_ at each time point of measurement, where R_n_ is the cell-electrode impedance of the well when it contains cells and *R*_b_ is the background impedance of the well with the medium alone.

### 2.7. Measurement of the Electrical Resistance of Caco-2 Cell Layers

For the measurement of barrier integrity Caco-2 cells were seeded onto culture inserts (Transwell 3460, polyester membrane, 0.4 µm pore size, Corning Costar) and cultured for three weeks [2,24]. Transepithelial electrical resistance (TEER) reflects the tightness of the intercellular junctions. TEER was measured by an EVOM volt-ohmmeter (World Precision Instruments, Sarasota, FL, USA) combined with STX-2 electrodes, and was expressed relative to the surface area of the monolayers (Ω × cm^2^). Resistance of cell-free inserts (130 Ω × cm^2^) was subtracted from the measured values. TEER values were measured before and right after permeability experiments). TEER values indicated the integrity and paracellular permeability of cell layers for ions. The TEER of Caco-2 monolayers was 1302 ± 49 Ω × cm^2^ (mean ± SD; *n* = 46) after 3 weeks culturing. For the recovery experiment, after changing the 10 μM peptide solutions to culture medium, TEER was monitored for one day.

### 2.8. Penetration of Marker and Drug Molecules across Caco-2 Cell Layers

In the permeability experiments inserts with Caco-2 cell layers grown for 3 weeks were transferred to 12-well plates containing 1.5 mL Ringer–Hepes buffer in the lower (basal or acceptor) compartments. In the upper or apical compartments culture medium was replaced by 0.5 mL buffer containing peptide solutions: PN159 peptide at 1, 3, 10 and 30 μM concentrations and CPPs at 100 μM concentration. Permeability marker molecules albumin (1 mg/mL; *M*w: 65 kDa) labeled with Evans blue (167.5 μg/mL) and fluorescein (10 μg/mL; *M*w: 376 Da) were added 30 min later. The incubation with permeability markers in the presence of PN159 peptide lasted for 30 min. In case of CPPs the incubation time with fluorescein was 1-h. Samples were collected from both compartments and the concentrations of the marker molecules were determined by a fluorescence multi-well plate reader (Fluostar Optima; excitation wavelength: 485 nm, emission wavelength: 535 nm in the case of fluorescein and excitation wavelength: 584 nm, emission wavelength: 680 nm in the case of Evans-blue labeled albumin). Fluorescein isothiocyanate (FITC)-labelled dextran marker molecules (Table 1) were used at 1 mM concentrations, and fluorescence intensities of collected samples were measured using a Fluorolog FL3-22 (Horiba Jobin Yvon, Paris, France) spectrofluorometer using 492 nm excitation wavelength, 515 nm emission wavelength, and 0.5 s integration time. Drugs were used at 10 μM concentrations and samples were measured by HPLC as described below. The apparent permeability coefficients (P_app_) were calculated as described previously [8,24]. Briefly, cleared volume was calculated from the concentration difference of the tracer in the basal compartments (Δ[C]_B_) after 30 min and apical compartments at 0 h ([C]_A_), the volume of the basal compartment (V_B_; 1.5 mL) and the surface area available for permeability (A; 1.1 cm^2^).

### 2.9. High-Performance Liquid Chromatography (HPLC)Analytical Procedures

Analytical measurements were performed on a Merck-Hitachi LaChrom HPLC system equipped with an ultraviolet (UV) and fluorescence detector (Merck, Darmstadt, Germany). All reagents used were of analytical reagent grade. Atenolol and verapamil were determined by using a Gemini C18 column (150 × 3 mm, 5 μm, 110 Å; Phenomenex Inc., Torrance, CA, USA) operated at 0.5 mL/min flow rate, maintained at 35 °C. For atenolol measurements isocratic elution was applied with a mixture of 300 mL methanol, 400 mL 100 mM ammonium acetate, 20 mL 10% ammonium-hydroxide and 2 mL 0.1 M ethylenediaminetetraacetic acid. The mobile phase for verapamil consisted of 350 mL methanol with 250 mL 20 mM ammonium acetate and 2 mL 10% trifluoroacetic acid with isocratic elution, at 0.5 mL/min flow rate. UV detection for atenolol was performed at 270 nm while for verapamil the 230 nm wavelength was adjusted. The calibration curve was linear (*r*^2^ = 0.9999) over a range of 0.01–1.5 μM for atenolol. The calibration curve for verapamil also showed a good linearity within the examined concentration range of 0.01–1 μM (*r*^2^ = 0.9952). In both cases 10μL sample was injected into the chromatographic system. Samples at higher than 1 μM atenolol and verapamil concentrations were 10× diluted with Ringer–Hepes solution.

HPLC measurement of quinidine was performed using a YMC Pack Pro C18 column (RS 150 × 2.1 mm, 5 μm, 80 Å; YMC America Inc., Allentown, PA, USA) equipped with a guard column operated at 0.2 mL/min flow rate. The chromatographic column was maintained at 40 °C and the 5 μL sample was injected onto the column. The elution of quinidine was performed with a buffer containing a mixture of 100 mL methanol, 250 mL 0.1% ammonium acetate, and 4 mL 10% trifluoroacetic acid. Quinidine was quantified at 250/430 nm excitation/emission wavelengths. The calibration curve was linear over the quantitation range of 0.01–1 μM (*r*^2^ = 0.9984).

HPLC measurement of cimetidine was performed using a Gemini NX column (150 × 4.6 mm, 3 µM) equipped with a guard column operated at 0.8 mL/min flow rate. The column temperature was maintained at 40 °C and a 20 μL sample was injected onto the chromatographic system. A buffer comprising of 15% acetonitrile and 85% 100 mM ammonium acetate was utilized for the isocratic elution of cimetidine. UV detection was set at 228 nm. The calibration curve was linear (*r*^2^ = 0.9999) over the range of 0.01–10 μM.

### 2.10. Electron Microscopy

Cells grown on culture inserts were treated with 10 μM PN159 peptide for 30 min and fixed immediately after treatment or after a 1-day recovery. Cells were briefly washed with phosphate-buffered saline (PBS) and fixed with 3% paraformaldehyde containing 0.5% glutaraldehyde in cacodylate buffer (pH 7.4) for 30 min at 4 °C. After washing with the buffer several times, cells were postfixed in 1% OsO_4_ for 20 min. Following a rinse with distilled water, the cells were block stained with 1% uranyl acetate in 50% ethanol for 20 min, dehydrated in graded ethanol and after the last step of dehydration inserts were placed in the 1:1 mixture of ethanol and Taab 812 (Taab; Aldermaston, Berks, UK) for 10 min at 30°C. Finally, the membranes of the culture inserts with the cells were removed from their support and embedded in Taab 812. Polymerization was performed overnight at 60°C. Ultrathin sections were cut perpendicularly for the membrane using a Leica EM UC6 ultramicrotome (Nussloch, Germany) and picked up on formvar-coated single-slot copper grids. The sections were examined using a Hitachi 7100 transmission electron microscope (Hitachi Ltd., Tokyo, Japan) and a side-mounted Veleta CCD camera (Olympus Soft Imaging Solutions). Altogether, 105 non-overlapping images from the 3 groups at the same magnification were analyzed for the presence or absence of tight junctions (control group *n* = 28; PN159 group *n* = 42; recovery group *n* = 35).

### 2.11. Immunohistochemistry

Morphological changes in interepithelial junctions after treatment with different concentrations of CPPs (100 μM) and PN159 peptide (1, 3, 10, 30 and 100 μM) were followed by immunostaining for claudin-1 integral membrane tight junction protein, β-catenin and zonula occludens protein-1 (ZO-1) cytoplasmic linker proteins. F-actin was stained by fluorescently labeled phalloidin. Treatments lasted for 1 h. After peptide treatment cell layers were washed with PBS and fixed with a 1:1 mixture of ice cold acetone and methanol for 2 min. Cells were blocked with 3% bovine serum albumin in PBS and incubated with primary antibodies mouse anti-claudin-1 and -4, rabbit anti-ZO-1 and rabbit anti-β-catenin (Life Technologies, Carlsbad, CA, USA) overnight. Fluorescent Atto 647N-phalloidin (ATTO-TEC GmbH, Germany) lasted for 1 h. Incubation with Alexa Fluor-488-labeled anti-mouse and anti-rabbit secondary antibodies (Life Technologies, Invitrogen, USA) lasted for 1 h. Bis-benzimide dye (Hoechst 33342) was used to stain cell nuclei. After mounting (Fluoromount-G; Southern Biotech, Birmingham, AL, USA) the samples staining was visualized by a Leica TCS SP5 confocal laser scanning microscope (Leica Microsystems GmbH, Wetzlar, Germany) and Visitron spinning disk confocal system (Visitron Systems GmbH, Puchheim, Germany).

### 2.12. Circular Dichroism (CD) Spectroscopy

Far-UV circular dichroism (CD) spectra of the peptides were recorded at 25°C temperature on a J-810 spectropolarimeter (JASCO International Co. Ltd., Tokyo, Japan). The CD spectra were measured between 260 nm and 185 nm with an optical pathlength of 1 mm, at a peptide concentration of 0.1 mg/mL in Milli-Q water diluted directly from powder. The bandwidth was 2 nm and data pitch 0.5 nm, the scan speed was set to 100 nm/min and the integration time was 1 sec. Ten spectra were accumulated and plotted. The data were analyzed by CDSSTR method [27]. To test the thermostability of the peptide structure the CD spectra were recorded between 25 °C and 55 °C temperature with a ramp rate of 5 °C/min on a JASCO J-810 spectropolarimeter by using a Peltier sample holder. The CD spectra were measured immediately between 260 and 185 nm with an optical pathlength of 1 mm, the peptide concentration was 0.1 mg/mL in Milli-Q water. The bandwidth was 2 nm and data pitch 1 nm, the scan speed was set to 100 nm/min, and the integration time was 1 sec. Five spectra were accumulated and plotted.

### 2.13. Molecular Modelling

Protein structures were obtained by homology modeling using the MODELLER program package [28]. First, the structure of mouse claudin-15 was completed by homology modelling because its X-ray structure file is lacking the fragment of residues from 34 to 41 [29]. Human claudins 1, 4 and 7 were homology modeled using the completed mouse claudin-15 as a template. The protein homology models were further relaxed by a short (400 ns) molecular dynamics simulation in which the proteins were embedded in a 95 by 95 Å POPC bilayer-water system containing 150 mM NaCl. Molecular dynamics simulations were performed by the program NAMD [30], using the CHARMM27 molecular force field with CMAP correction. The results were visualized by VMD v1.9.1. [31]. The docking studies were performed by the CABS docking server [32] and the resulting C-alpha traces were reconstructed by MODELLER using the python script supplied by the server homepage.

### 2.14. Visualization of the Uptake of PN159 Peptide in Caco-2 Cells

The Caco-2 cells were grown on glass bottomed petri dishes coated with collagen to visualize the cellular uptake of the Bodipy FL maleimide labeled (BODIPY FL *N*-(2-aminoethyl) maleimide, Thermo Fischer, Waltham, MA, USA) PN159 peptide. The cells were treated with peptide at concentrations of 1, 3 and 10 µM for 5 min. To stain the cell nuclei the H33342dye (1 μg/mL) was used before peptide treatment for 10 min. After peptide incubation the living cells were washed three times with Ringer–Hepes buffer supplemented with 1%fetal bovine serum (FBS) and examined immediately with a Visitron spinning disk confocal system.

In the case of the co-localization analysis the labeled PN159 peptide was used in 10 µM for 5 min. Then the cellular uptake cells were fixed and immunohistochemistry was performed for claudin-4. The samples were mounted and visualized by a Leica TCS SP5 confocal laser-scanning microscope.

### 2.15. Determination of Minimum Inhibitory Concentration (MIC) on Microbial Pathogens

We tested the spectrum of activity of PN159 on a set of five sensitive ESKAPE pathogens: *Staphylococcus aureus* (ATCC 29213), *Klebsiella pneumoniae* (ATCC 10031), *Acinetobacter baumannii* (ATCC 17978), *Pseudomonas aeruginosa* (ATCC 27853), *Enterobacter cloacae spp. cloacae* (ATCC 13047); and one vancomycin resistant ESKAPE pathogen, *Enterococcus faecium* (ATCC 700221). The MIC of PN159 on the 6 sensitive and resistant ESKAPE pathogenic strains was determined in cation-adjusted Muller–Hinton broth. For *Enterococcus faecium*, which does not grow in Muller-Hinton broth, Brain-Heart-Infusion broth was used. Minimum inhibitory concentrations (MICs) were determined by using a standard serial broth dilution technique [33]. Briefly, 11-step serial dilutions were prepared in 96-well microtiter plates with three biological replicates per strain. Pathogens were inoculated into each well at a density of 5×10^5^ bacteria/mL, and the plates were incubated at 37 °C. Plates were shaken at 300 rpm during incubation for 18 h. Cell growth was monitored by measuring the optical density (OD600 value, Synergy 2 microplate reader BioTek Instruments Inc, Winooski, VT, USA). MIC was defined as complete growth inhibition (i.e., OD600 < 0.05). As reference compounds we tested the antimicrobial activity of three bactericidal drugs with different modes of action. Cefoxitin inhibits the cell wall synthesis, gentamicin (Applichem GmbH, Darmstadt, Germany) is a 30S ribosomal subunit inhibitor, while ciprofloxacin is a gyrase inhibitor. The highest tested concentrations were 100 µg/mL in the case of cefoxitin and gentamicin, and 10 µg/mL for ciprofloxacin.

### 2.16. Statistical Analysis

For statistical analysis Graph Pad Prism 5.0 software (Graph Pad Software Inc., San Diego, CA, USA) was used. All data presented are means ± SD. Values were compared using analysis of variance followed by Dunnett’s test. Changes were considered statistically significant at *p* < 0.05. All measurements were repeated three times and the number of parallel samples was minimum three.

## 3. Results

### 3.1. Concentration-Dependent Effect of PN159 Peptide on Epithelial Cell Viability

The colorimetric endpoint MTT test was performed after a 1-h treatment with different PN159 peptide concentrations at three different time points: (i) immediately after the 1-h treatment, (ii) at one-day recovery and (iii) at one-week recovery (Figure 1A). Low concentrations of the peptide (1–10 μM) did not decrease cell viability, while cell damage was found at higher, 30 and 100 μM concentrations. The cytotoxic effect of PN159 at 100 μM concentration was not reversible after one day or even one week. The kinetics of PN159 effects on Caco-2 cells were followed by real-time impedance measurements after 1-h (Figure 1B) or one-day treatment (Figure 1C). In both conditions only the two highest peptide concentrations decreased the cell impedance indicating cell damage, similarly to the results of the MTT assay.

Based on both tests, PN159 peptide treatment for 1-h was non-toxic at 10 μM or lower concentrations, reversible at 30 μM and toxic at the highest 100 μM concentration. The results obtained by MTT test and impedance measurement on Caco-2 cells were similar (Figure 1) and indicated that all the used concentrations of PN159 peptide were safe, except the 100 μM. Because of its toxicity, the 100 μM concentration was not used in further experiments.

### 3.2. Concentration-Dependent Effect of PN159 Peptide on Intestinal Epithelial Barrier Integrity

All tested concentrations of the PN159 peptide significantly decreased the resistance of epithelial cell layers after a 1-h treatment (Figure 2A). The two highest concentrations, 10 and 30 μM, opened the paracellular barrier the most, causing 80–90% decrease in TEER. In concordance, these two highest concentrations of PN159 showed the most effective permeability enhancer activity for both fluorescein and albumin (Figure 2B). Because the peptide was safe for the cells but effectively opened the barrier at 10 μM, this concentration was selected to reveal the kinetics of the reversibility of barrier opening, and to investigate its effects on drug penetration.

### 3.3. Reversible Effect of PN159 Peptide on the Opening of the Paracellular Cleft

The effect of PN159 peptide was very rapid, the electrical resistance dropped already to 42% of the control value after 1-min treatment (Figure 3A). The decrease of TEER continued until the end of the 30-min treatment (5 min: 16%, 15 min: 2.7%, 30 min: 1.4% of the control value). After the removal of the peptide the barrier function of Caco-2 cells was restored about 44% of the control value at six hours and a complete recovery could be observed at the 20 h timepoint. Intact TJs providing the morphological basis of barrier functions were visualized between Caco-2 epithelial cells in the control group by transmission electron microscopy, but no junctions were observed following treatment with PN159 peptide (Figure 3B). The disappearance of intercellular TJs was reversible, because after 1-day recovery the ultrastructure of epithelial junctions became similar to control cells. No open TJs were observed in the control or recovery groups by checking the electron micrographs (28–42 images/groups).

### 3.4. Effects of PN159 Peptide on the Penetration of Dextran Marker Molecules and Drugs

All these previous functional and morphological results pointed to the TJ opening effect of PN159 in Caco-2 cells which potentially can be exploited to increase drug penetration across the intestinal barrier. The permeability of Caco-2 monolayers was measured in the apical to basal (intestine to blood) direction for four water-soluble dextran marker molecules of different sizes (4–40 kDa) and four drugs, the hydrophilic atenolol and cimetidine, and the lipophilic quinidine and verapamil (Table 2). All four drugs are substrates of active efflux transporters [24]. The apparent permeability coefficients of the large fluorescein isothiocyanate-labeled dextran (FD) macromolecules were very low in control conditions but they were elevated by 159-400 fold following PN159 treatment. The highest increase was measured in the case of FD-40, the largest macromolecule. In control conditions the permeability of atenolol and cimetidine was the lowest from the tested molecules, while the highest penetration was measured for quinidine and verapamil on Caco-2 cells. PN159 treatment caused more than 30-fold change for atenolol and cimetidine which penetrate slowly across the cells layers. The peptide increased about 2 fold the permeability of the intestinal culture model for the lipophilic quinidine and verapamil, which already showed a good penetration.

### 3.5. The Effect of PN159 Peptide on the Staining of Junctional Proteins and F-Actin

The concentration-dependent effects of the peptide on epithelial barrier integrity in Caco-2 cells were confirmed by the immunostaining of junctional proteins claudin-1, ZO-1 and β-catenin, and the labeling of the actin cytoskeleton (F-actin) (Figure 4). The 1 μM concentration of the peptide caused a slight effect on cell morphology. At higher than 3 μM concentrations significant changes were observed both in the junctional protein pattern and in the actin cytoskeleton organization. Claudin-1 was the junctional protein most sensitive to the peptide treatment. PN159 at a concentration of 10 μM caused a drastic change in actin cytoskeleton and epithelial junctional morphology, with a visible opening of intercellular junctions.

### 3.6. Molecular Modeling of Human Claudin Proteins and Docking of PN159 Peptide

According to the CD spectra obtained, the secondary structure of PN159 peptide contains 11% α-helix, 31% β-sheet, 24% turns and 34% unordered structure (Figure 5A). This secondary structure of PN159 peptide was stable between 25 and 55 °C (Figure 5B). In concordance with the result of CD spectra, similar secondary structure motifs were found by molecular modeling (Figure 5C).

Claudin protein structures were obtained by homology modeling using the MODELLER program package (Figure 6A). Docking of PN159 peptide to full length homology modeled human claudin-1, -3, -4, and -7 monomers, highly expressed in Caco-2 cells [34] was performed on the CABS server (Figure 6B). Favorable docking poses located around extracellular loops ECL1 and ECL2 were sought by both energetic and geometric considerations and analyzed in all docking trajectories. Docking energies (“total energy”) were decomposed to “ligand energy”, “interaction energy” and “receptor energy” parts (Table 3). Correct docking poses were expected to have low values not only for the total energies, but as low as possible for all energy components simultaneously. Thus, the “ligand energy”, “interaction energy” and “total energy” values were investigated. Based on the modeling, energetically highly favorable interactions were found between PN159 and the ECLs of claudin-1, -4 and -7, but not for claudin-3 (Table 3). Docked poses of PN159 with ECLs of claudin-1, -4 and -7 are shown on Figure 6B.

The amino acid sequence of ECL1 and ECL2 of the three claudins which showed interaction with the TJ modulator peptide were compared and the most important amino acids which interact with PN159 were identified (Figure 6C). Four residues in ECL1, the polar Q44 and S/N/D53, the hydrophobic V55 and L71, and in ECL2 the polar N/Q156 participate in major interactions with the peptide. Q44 and V55 residues of claudin-1 and -7 form a binding pocket for L11 of the peptide, and the same two residues of claudin-4 bind L6 and K9 (Figure 6C). The polar S/N/D53 and L71 of claudin-4 and -7 interact with K5, while the same amino acids of claudin-1 bind K12. The N/Q156 amino acid of ECL2 of all three claudins interacts with lysine: in the case of claudin-1 this is L17 of PN159, for claudin-4 L2, and for claudin-7 L6, L8, L15, L17. Based on the docking results, lysine, especially K1 and K5 and leucine L6 and L17 of PN159 interact with all three claudins examined (Figure 6D). After docking of the peptide disappearance of β-strands in ECLs of claudins was observed (Figure 6B), which seems to be correlated with the strength of interaction (Table 3). Comparing the docking energy values of PN159 to claudins the following rank order in the strength of interaction can be established: claudin-3 < claudin-1 < claudin-4< claudin-7. The energy values from modeling are shown in Table 3.

### 3.7. Cell-Penetrating Effect and Uptake of PN159 Peptide in Epithelial Cells

Besides the junctional effects, changes in the cell membrane integrity were also examined by ethidium homodimer-1 (856.77 Da) staining of Caco-2 epithelial cells after 24-h treatments in the concentration range of 1 to 10 μM. Only the highest PN159 concentration caused plasma membrane entry of the red fluorescent dye, which stained cell nuclei (Figure 7A).

The uptake of the fluorescently labeled PN159 peptide, reflected by the green fluorescent signal, was visualized by confocal microscopy (Figure 7B). In Caco-2 cells treated with 1 and 3 μM concentrations, the peptide was detectable in the cytoplasm as green dots. The cellular uptake of the peptide in cells incubated with 10 μM concentration was more pronounced.

We confirmed the interaction of PN159 peptide with claudin-4 junctional protein by a co-localization analysis. The labeled peptide was visible in the cytoplasm and also in the cell membrane at the level of intercellular junctions immunolabeled for claudin-4 (Figure 8).

### 3.8. Antimicrobial Effect of PN159

We tested the antimicrobial activity of PN159 peptide on six ESKAPE pathogens (Table 4) at a wide range of concentration (0.8 to 70 µM). Among the tested bacteria *Acinetobacter baumannii* and *Enterococcus faecium* were the two most sensitive strains with MIC values below 5 µM. The concentration of PN159 to inhibit the growth of *Staphylococcus aureus* and *Klebsiella pneumoniae* was around 10 µM, which was in similar range where peptide effects were seen on human epithelial cells. The peptide was still effective on the vancomycin-resistant *Enterobacter cloacae* and on *Pseudomonas aeruginosa*, but at 10-times higher concentrations than in the case of the most sensitive pathogens.

As compared to the reference bactericidal antibiotic cefoxitin, PN159 was more effective in the case of *Acinetobacter baumannii*, *Enterococcus faecium*, *Enterobacter cloacae* and *Pseudomonas aeruginosa*. Gentamicin and ciprofloxacin showed strong antimicrobial activity for all ESKAPE pathogens, except *Enterococcus faecium*. This strain was resistant for all investigated antibiotics, however, PN159 efficiently inhibited its growth (Table 4).

### 3.9. The Effect of Cell-Penetrating Peptides on Intestinal Barrier Integrity

The kinetics of the effects of three other CPPs on Caco-2 cells were followed by real-time impedance measurements (Figure 9A). The R8 and φ-peptides decreased the cell impedance during the 24-h monitoring which indicates a cell reaction. Changes in cell index without cytotoxic effects most probably are caused by an increase in plasma membrane ionic permeability. There was no major influence of CPPs on barrier integrity (Figure 9B,C) as compared to the effect of PN159. We observed a slightly decreased TEER in the case of R8 and φ-peptide and only the φ-peptide caused an elevated permeability for the fluorescein marker molecule. Based on these results the investigated CPPs have no significant opening effect on the intercellular junctions, which was also verified by immunostaining for junctional proteins (Figure 9D).

## 4. Discussion

We investigated for the first time the dual, TJ modulator and cell membrane actions of the PN159 peptide. PN159 was described as a TJ modulator and permeability-enhancing peptide on airway epithelial cell layers in vitro and in vivo [4], and in our previous comparative work we found it also very effective on culture models of the blood–brain and intestinal barriers [8]. The same peptide, known also as KLAL or MAP was extensively studied for its cell membrane action as a CPP [10,11,12,35]. However, the identity of the differently named molecule was not previously known for the researchers of the two fields. We demonstrated, that beside the cell-penetrating and antimicrobial effects, PN159 is a TJ modulator on intestinal epithelial cells and by molecular modeling new members of the claudin family were identified as its potential targets.

### 4.1. TJ Modulator Effect of PN159: Safety, Concentration Dependence and Reversibility

All the presented functional and morphological results point to the safe, concentration-dependent and reversible TJ opening effect of PN159 in Caco-2 cells. The effect of the peptide on epithelial cell viability was investigated for one and 24 h. We found that the peptide has no effect on cell metabolism measured by MTT test or on impedance kinetics and can be considered as safe up to 10 μM. The changes seen after higher, 30 μM concentration were fully reversible. The 100 μM concentration of PN159 caused irreversible changes in both parameters on intestinal epithelial cells after either 1-h or one-day treatment. Using respiratory epithelial tissue (EpiAirway model) and MTT assay, PN159 was found to be safe at concentrations up to 100 μM for a 1-h treatment [9]. The difference between the sensitivity of the epithelial models and also the sensitivity of the applied methods may explain the dissimilarity of these data.

We found a marked concentration-dependence of the TJ modulator effect. With the most sensitive method, TEER, we could determine that 1 μM PN159 already opened the paracellular cleft for ions, while higher concentrations gradually decreased TEER. This TJ modulation by the effective but safe 10 μM concentration of the peptide was fast, within minutes, but after removal of the peptide a full recovery was seen. This opening and complete recovery of the intestinal TJ structure was verified by electron microscopy in the present study and by immunohistochemistry for TJ proteins both in this and our previous work [8]. A similar concentration-dependent and reversible effect of PN159 on TEER was found on airway epithelial tissue, too [4].

The concentration dependence of the TJ modulator effect of the peptide was also detected in the permeability assay for a small and a large hydrophilic marker, fluorescein and albumin. A similarly high permeability increase with a full reversibility was measured for the same markers on both Caco-2 and blood–brain barrier culture models with the 10 μM concentration of PN159 peptide [8]. This action of the peptide was verified by the immunostaining of junctional proteins with higher concentrations causing bigger alterations in junctional protein intensity and distribution. Other claudin-specific TJ modulator peptides act similarly on junctional protein staining as well as on permeability of epithelial and endothelial cells [36,37].

In addition, the opening of the paracellular pathway in Caco-2 monolayers was measured by four dextran markers with size between 4 and 40 kDa. The apparent permeability coefficients of the large dextran macromolecules were very low in control conditions. PN159 treatment (10 μM) elevated the dextran penetration by 159- to 400-fold. The highest increase was measured in the case of the largest, FD-40 marker. The concentration dependency of the permeability increasing effect of PN159 was also observed on the EpiAirway model, but in the concentration range of 25 to 100 μM. The elevation of the permeability was 10- to 30-fold for FD-4 [4] and for peptide hormones with the same size [9].

To study the effect of PN159 on drug penetration, we tested two hydrophilic and two lipophilic drugs, which are substrates of active efflux transporters [24,38]. The permeability of hydrophilic atenolol and cimetidine was very low, while high penetration was measured for the lipophilic quinidine and verapamil on Caco-2 cells in agreement with our previous study [24]. PN159 treatment caused more than 30-fold permeability elevation for atenolol and cimetidine which penetrate slowly across the cell’s layers. The peptide increased about 2-fold the permeability of the intestinal culture model for the lipophilic quinidine and verapamil, which already showed a high penetration. The permeability of the slightly lipophilic plant alkaloid, galantamine (287 Da), a substrate of P-glycoprotein, was also increased 2 to 3-fold by PN159 in airway epithelial tissue model [9] confirming the peptide’s penetration enhancer effect.

### 4.2. Tight Junction (TJ)Modulator Effect of PN159: Interaction with Claudins

In our previous study, binding of PN159 in the micromolar range to epithelial claudin-1 and endothelial claudin-5 was demonstrated by affinity measurements and confirmed by docking studies [8]. In this report we describe an additional two intestinal epithelial junctional proteins, claudin-4 and -7 which are highly expressed in Caco-2 cells [34], as potential targets of the peptide by molecular docking. All these results suggest that PN159 peptide may open cell–cell junctions by acting on claudins, the most prominent family of integral membrane junctional proteins. C-CPE and C1C2 tight junction modulator peptides were also described to target claudins. C-CPE is a fragment of *Clostridium perfringens* enterotoxin which directly binds claudin subtypes including claudin-3 and -4 and induces disintegration of tight junctions [39]. The peptidomimetic C1C2 which contains the C-terminal half of ECL1 of claudin-1 predominantly binds to claudin-1 and -5, opens the paracellular barrier in cultured cells and causes cytosolic distribution of claudin-1, -2, -3, -4, and -5 [36]. What is common in the action of these three TJ modulator peptides is that they have specific interactions with conserved amino acids of claudins [8,40], however, these peptides do not share any similarity in their amino acid sequence. The secondary structure of C1C2 consists of β-sheet stabilized by α-helix [40], the claudin-binding domain of C-CPE is a nine-stranded β-sandwich [41], while according to our CD measurements and molecular modelling PN159 also contains β-sheet in addition to α-helix. This structural similarity between the three different but claudin-targeting peptides may be linked to the TJ modulator effect.

### 4.3. Cell Penetration and Antimicrobial Effect of PN159

We observed a concentration-dependent cellular uptake of PN159 on intestinal barrier cells as reflected in the confocal microscope images. PN159 has a fast uptake in human melanoma cells, treatment with 1 µM concentration already causing detectable entry of the peptide into the cells, in concordance with our results [12]. The cellular uptake of PN159 was also verified on other four cell lines [42]. After treatment with a concentration of 10 µM, the peptide entered the cytoplasm of cells without cytotoxic effects, similar to our data.

Most of the CPPs have antimicrobial activity due to their structural properties, especially the cationic and the amphipathic membrane active peptides [13]. Based on literature data the amphipathic PN159 effectively inhibited the growth of different microorganisms, like *Escherichia coli, Staphylococcus epidermidis, Bacillus megaterium, Saccharomyces cerevisiae* [10,19,43]. In this study we demonstrated the antimicrobial activity of PN159 peptide in the concentration range of 3–46 µM on clinically relevant ESKAPE pathogens. As compared to the reference antibiotics, PN159 was more effective in the inhibition of the growth of *Acinetobacter baumannii*, *Enterococcus faecium*, *Enterobacter cloacae* and *Pseudomonas aeruginosa*. (Table 4).

Comparing our results with data from the literature, another amphipathic antimicrobial peptide, melittin, inhibited the growth of *Escherichia coli* and *Micrococcus luteus* [44] and was also tested against three of the ESKAPE pathogens. *Acinetobacter baumannii* and *Pseudomonas aeruginosa* were more sensitive for melittin treatment, than *Klebsiella pneumoniae:* for the inhibition of growth of *Klebsiella pneumoniae* more than 100 µM melittin was needed [45], while in our study around 10 µM PN159 was already effective.

PN159, as a model amphipathic CPP peptide, strongly interacts with biological membranes due to its physical-chemical properties. As a CPP, PN159 enters epithelial cells [42], enhances the membrane penetration of molecules and exerts antimicrobial effects [10,11]. Because PN159 binds different claudins and interacts specifically with their conserved amino acids, it acts as a TJ modulator [8]. It is an open question, however, if the plasma membrane and the TJ modulator actions of PN159 are connected or not.

### 4.4. TJ Modulator Effect of Other CPPs?

Based on these results we asked the question whether this TJ modulator ability is a unique effect of PN159 among the CPPs or not. Therefore, we have tested two other amphipathic CPPs, the widely used Pep-1 [16] and the cyclic φ-peptide [15], and one cationic CPP, R8 peptide [18] on the barrier integrity of Caco-2 monolayers. None of the tested CPPs had a TJ modulator effect similar to PN159. There was no change in the TEER values and only the φ-peptide increased the permeability of fluorescein. Since no effect was seen on junctional staining in epithelial cells treated with φ-peptide, the increase in permeability and decrease in the cell layer impedance may be linked to the cell membrane action of this CPP [15]. As we demonstrated by our modeling results, PN159 peptide interacts with conserved residues of the claudins investigated (Figure 6) suggesting its ability for specific interactions [8]. The amino acid sequence of PN159 results in an amphipathic helix structure [10], which is not characteristic of the other CPPs investigated in our paper. From the other three investigated peptides, R8 and φ-peptide have positively charged arginine clusters (Table 1) and are considered as clearly cationic CPPs [17]. Pep-1 also contains a positively charged amino acid cluster and is rich in tryptophan repeats (Table 1). Neither PN159 contains clusters of arginine or positively charged amino acids, nor tryptophan (Table 1). While we have not found a CPP with a TJ modulator effect in the current study, we cannot exclude that other membrane active peptides can have such dual action. The amphipathic antimicrobial peptide melittin enhanced the paracellular permeability for dextran marker molecules on Caco-2 cells [46]. The opening of TJs by melittin was mediated, at least partially, by the prostaglandin signaling pathway [46].

## 5. Conclusions

We demonstrated that PN159 (KLAL/MAP) peptide as a penetration enhancer has a dual action on intestinal epithelial cells. The peptide safely and reversibly enhances the permeability of model molecules and drugs through intestinal epithelial cell layers by opening the intercellular junctions. We identified claudin-4 and -7 junctional proteins by docking studies as potential binding partners and targets of the peptide in the opening of the paracellular pathway. In addition to the tight junction modulator action, the peptide has a concentration-dependent cellular uptake and cell membrane permeabilizing effect, which can also contribute to its penetration-enhancing effect. This dual action is not general for cell-penetrating peptides, since the other three CPPs tested did not show barrier opening effects. Therefore, we propose the investigation of CPPs for testing their effect on paracellular permeability.

## Figures and Tables

**Figure 1 pharmaceutics-11-00073-f001:**
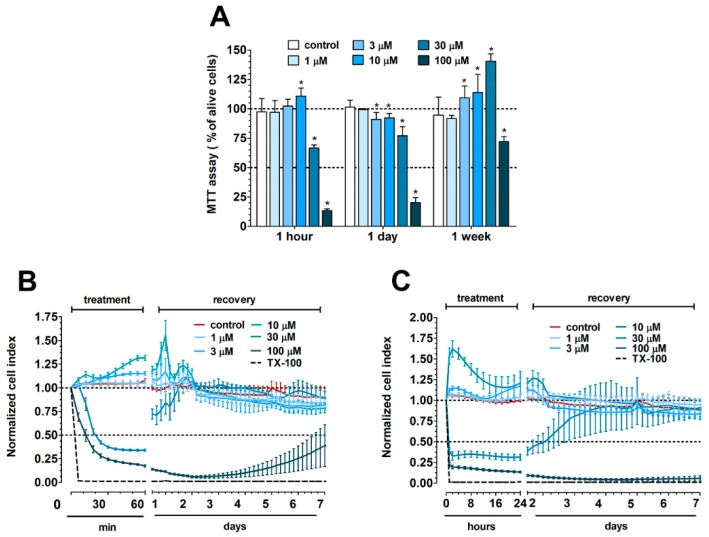
(**A**) MTT assay after 1-h treatment with PN159 peptide on Caco-2 cells followed by 1-h, 1-day or 1-week recovery. The MTT values are given as percent of the control group (100% viability). Values are presented as means ± SD, *n* = 3–8. Statistical analysis: analysis of variance (ANOVA) followed by Dunnett test, *p* < 0.05 as compared with the control groups. (**B**) Impedance measurements after 1-h treatment and (**C**) 1-day treatment followed by a recovery phase of 1-week. The effects of PN159 peptide on the impedance were shown as normalized cell index. Values are presented as means ± SD, *n* = 3–8.

**Figure 2 pharmaceutics-11-00073-f002:**
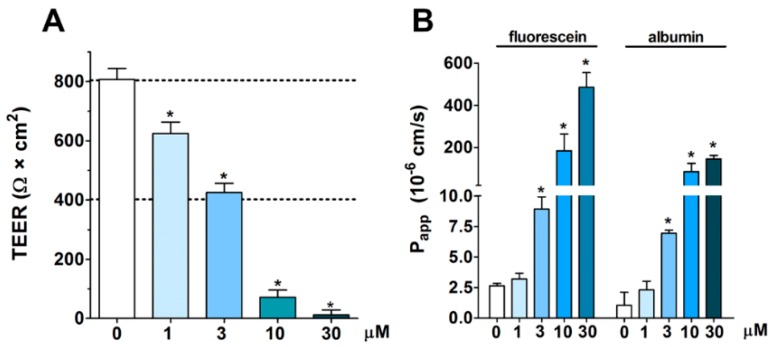
Effect of 1-h PN159 treatment on barrier integrity of Caco-2 cell layers. (**A**) Transepithelial electrical resistance (TEER). (**B**) Permeability for fluorescein and albumin marker molecules (P_app_ A-B 10^−6^ cm/s). Values are presented as means ± SD, *n* = 3–8. Statistical analysis: ANOVA followed by Dunnett’s test, *p* < 0.05 as compared with the control groups.

**Figure 3 pharmaceutics-11-00073-f003:**
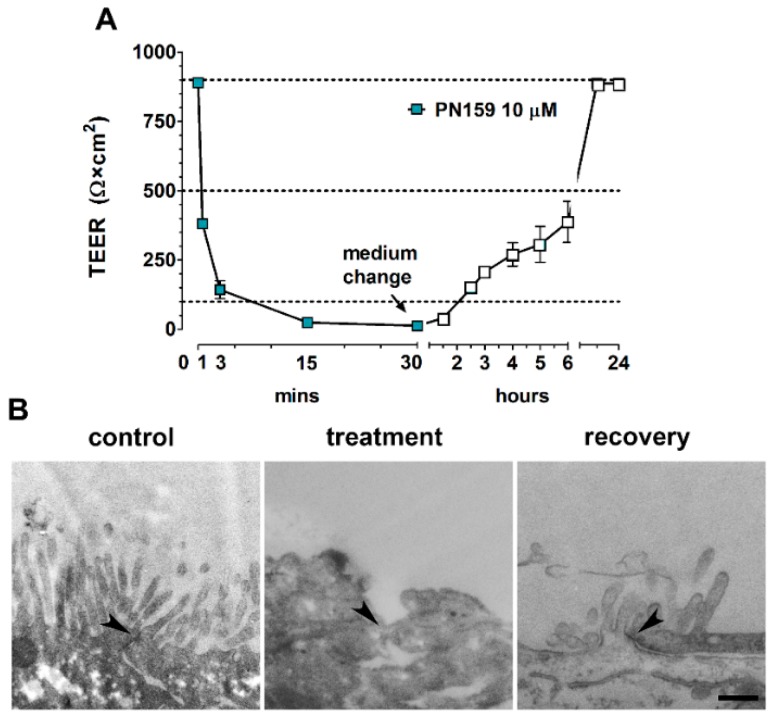
Reversible effect of PN159 peptide (10 μM, 30-min treatment, 24-h recovery) on Caco-2 cells. (**A**) Kinetic analysis of transepithelial electrical resistance (TEER) after PN159 peptide treatment. (**B**) Transmission electron micrographs of cell–cell connections (arrowheads); bar = 400 nm.

**Figure 4 pharmaceutics-11-00073-f004:**
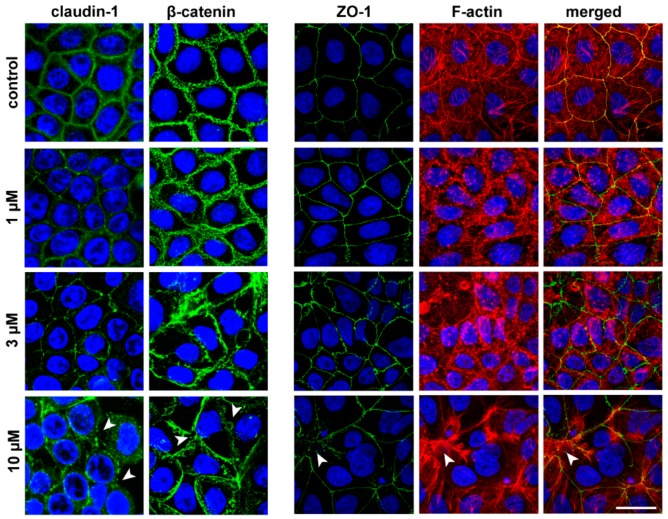
Effects of PN159 peptide on junctional morphology of Caco-2 cells. Immunostaining for claudin-1, β-catenin and zonula occludens-1 (ZO-1) junctional proteins and fluorescent staining for F-actin are shown in control conditions or after 1-hour peptide treatment. Green color: immunostaining for ZO-1, β-catenin and claudin-1. Blue color: staining of cell nuclei. Red color staining of F-actin. Bar = 20 µm. Arrowheads indicate the opened junctions between the cells.

**Figure 5 pharmaceutics-11-00073-f005:**
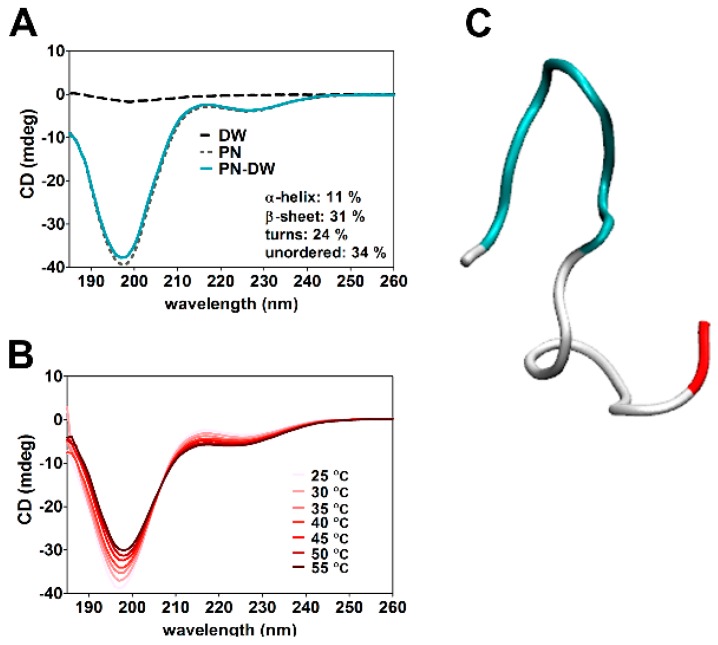
(**A**) Circular dichroism (CD) spectroscopy of PN159 peptide at 25 °C. (**B**) Thermostability measurement of the peptide structure. (**C**) Structure of PN159 by molecular modeling. Blue color: turn motifs; White color: coil-coil motifs; Red color: C-terminal region.

**Figure 6 pharmaceutics-11-00073-f006:**
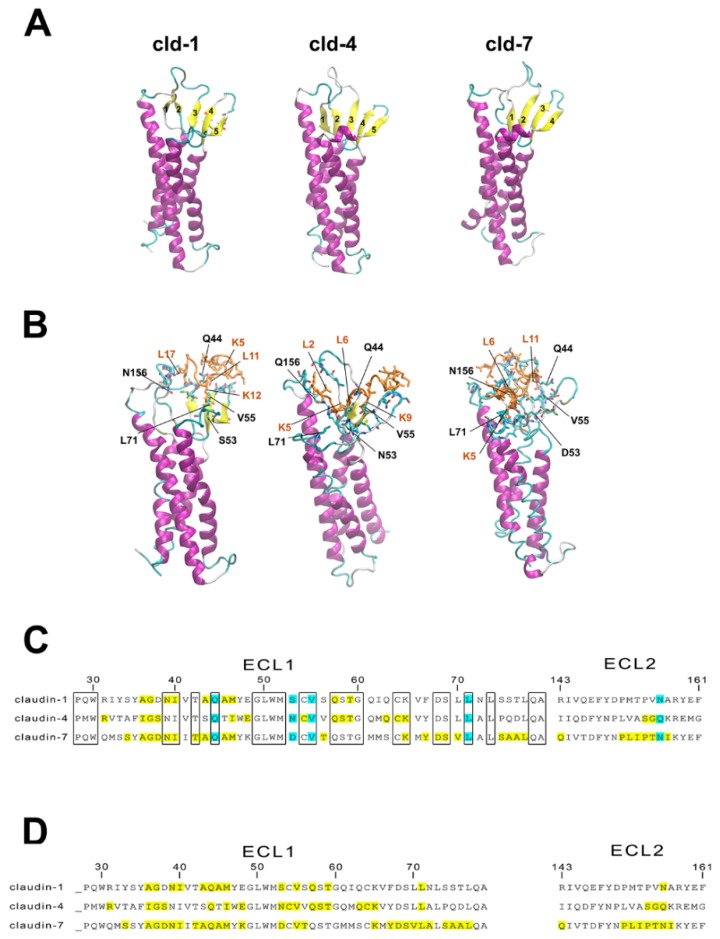
(**A**) Modeling human claudin-1, -4, -7 proteins. (**B**) Docking of PN159 peptide to claudin proteins. Interacting residues are shown as sticks. Orange color: PN159 peptide; Yellow color: β-strands; purple color: α-helices of claudins. (**C**) Amino acid sequence of ECL1 and ECL2 of human claudin monomers 1, 4 and 7. Conserved amino acids are indicated in the boxes. The amino acids which interact with PN159 according to docking studies are marked by yellow and light blue. Blue color indicates interaction between the amino acids of claudins and PN159 peptide at the same position in all three claudins. ECL: extracellular loop. (**D**) Interaction between PN159 peptide and human claudin monomers 1, 4 and 7. Amino acids of PN159 peptide which interact with the claudins according to docking.

**Figure 7 pharmaceutics-11-00073-f007:**
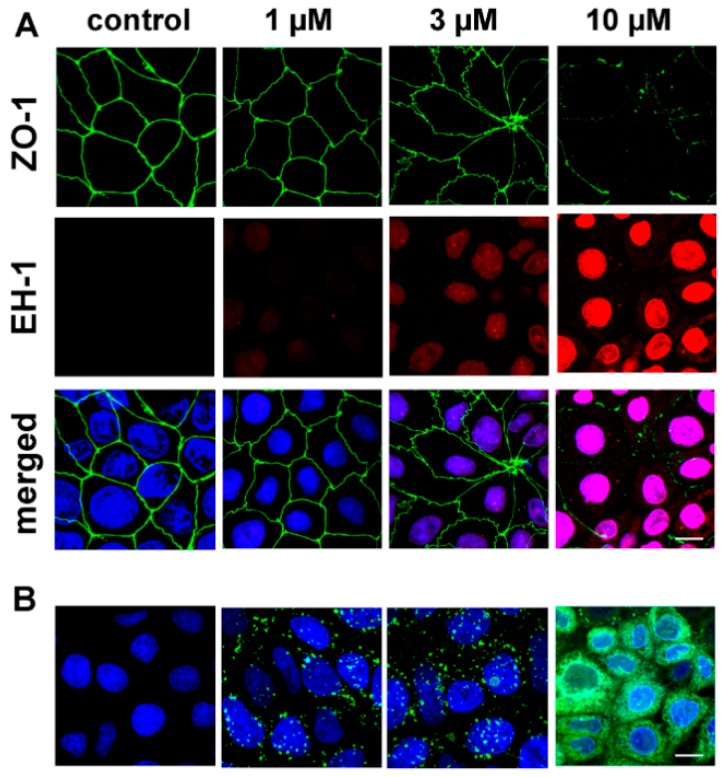
(**A**) Fluorescent staining of juntional protein zonula occludens-1 and double staining of cell nuclei in Caco-2 cells after 1-h treatment with PN159 peptide. Green color: immunostaining for ZO-1. Red color: staining by ethidium homodimer-1 (EH-1), indicating increased membrane permeability.Blue color: staining of all cell nuclei by H33342. Bar = 10 µm. (**B**) Confocal microscopy images of living Caco-2 epithelial cells incubated with Bodipy FL labeled PN159 peptide in different concentrations for 5 min at 37°C. Green color: BodipyFL-PN159 peptide. Blue color: staining of cell nuclei. Bar = 10 µm.

**Figure 8 pharmaceutics-11-00073-f008:**
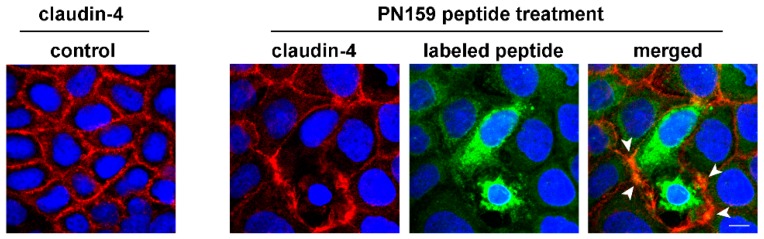
Caco-2 cells were treated with Bodipy FL labeled PN159 peptide (5 min), then fluorescent immunostaining of junctional protein claudin-4 and staining of cell nuclei were performed. Red color: immunostaining for claudin-4. Green color: Bodipy FL labeled PN159 peptide. Blue color: staining of cell nuclei by H33342. Bar = 10 µm. Arrowheads indicate co-localization of claudin-4 and the labeled PN159 peptide.

**Figure 9 pharmaceutics-11-00073-f009:**
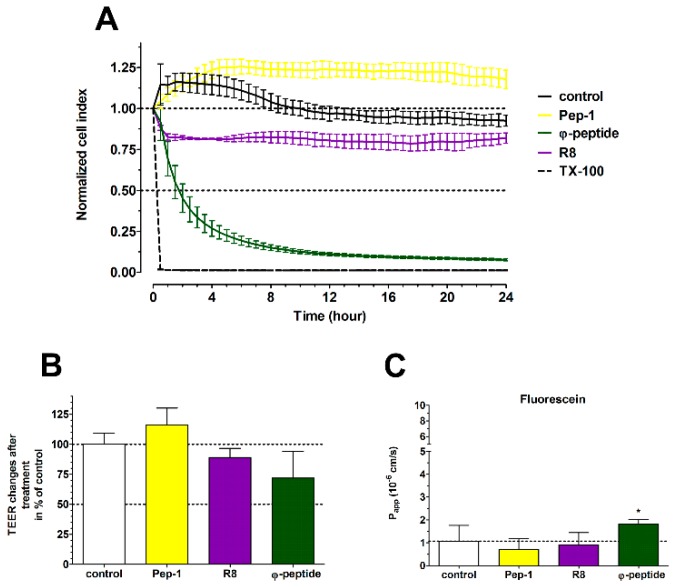
(**A**) Effects of different cell-penetrating peptides (CPPs; 100 μM, 24 h) on Caco-2 cells followed by impedance measurements. The effects of peptides on the impedance were shown as normalized cell index. Values are presented as means ± SD, *n* = 3−8. (**B**) Evaluation of barrier integrity on Caco-2 cell layers by measurement of transepithelial electrical resistance (TEER) after 1-h CPP treatment. (**C**) Caco-2 cell layer permeability for fluorescein marker molecule (P_app_ A-B 10^−6^ cm/s) after 1-h CPP treatment. (**D**) Effects of CPPs on junctional morphology of Caco-2 cells. Immunostaining for zonula occludens-1 (ZO-1) and β-catenin junctional proteins in control conditions and after 1-h peptide treatment. Red color: immunostaining for ZO-1 and β-catenin. Blue color: staining of cell nuclei. Bar = 20 µm.

**Table 1 pharmaceutics-11-00073-t001:** The amino acid sequences of the tested cell-penetrating peptides.

CPPs	Amino Acid Sequence	References
PN159 (KLAL/MAP)	KLALKLALKALKAALKLA-amide	[4,10]
Pep-1	KETWWETWWTEWSQPKKKRKV-amide	[16]
R8	RRRRRRRR-amide	[18]
φ-peptide	cyclo[CGGFWRRRRGE(εAca)G]	[15]

**Table 2 pharmaceutics-11-00073-t002:** Apparent permeability coefficients of dextran marker molecules and drugs in the apical-basal direction (P_app_ A-B, 10^−6^ cm/s) in control and PN159 treated cultures. The differences between control and treated groups were expressed in fold changes. FD, fluorescein isothiocyanate-labeled dextran.

Markers and Drug	Control	PN159	Fold Change
FD-4	0.021 ± 0.001	4.2 ± 0.1	200.0
FD-10	0.025 ± 0.010	4.8 ± 0.7	192.0
FD-20	0.022 ± 0.002	3.5 ± 1.0	159.0
FD-40	0.015 ± 0.003	6.0 ± 0.3	400.0
atenolol	1.2 ± 0.3	36.3 ± 1.9	30.0
cimetidine	0.9 ± 0.2	34.5 ± 4.5	38.3
quinidine	45.0 ± 13.3	72.6 ± 2.4	1.6
verapamil	46.2 ± 3.9	86.7 ± 18.8	1.9

**Table 3 pharmaceutics-11-00073-t003:** Docking energy components of PN159 to selected human claudin monomers. *E*_tot_, total energy; *E*_lig_, ligand energy; *E*_int_, interaction energy.

Energy	Claudin-1	Claudin-3	Claudin-4	Claudin-7
*E* _lig_	−34	−17	−61	−32
*E* _int_	−63	−59	−79	−186
*E* _tot_	−1228	−1163	−1229	−1798

**Table 4 pharmaceutics-11-00073-t004:** Antibacterial activity of PN159 peptide, cefoxitin, gentamicin and ciprofloxacin antibiotics on ESKAPE pathogens. ATCC numbers are added in brackets. MIC (µM), minimal inhibitory concentration.

ESKAPE Pethogens (ATCC)	PN159	Cefoxitin	Gentamicin	Ciprofloxacin
*Acinetobacter baumannii* (17978)	3.6	222.6	0.6	0.94
*Enterococcus faecium* (700221)	4.4	>222.6	>143.9	>30.18
*Staphylococcus aureus* (29213)	9.2	6.9	1.1	1.89
*Klebsiella pneumonia* (10031)	13.8	3.5	0.6	0.06
*Enterobacter cloacae* (13047)	31.0	>222.6	1.1	0.06
*Pseudomonas aeruginosa* (27853)	46.4	>222.6	2.3	1.89

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
