# Peer review of "Dual Action of the PN159/KLAL/MAP Peptide: Increase of Drug Penetration across Caco-2 Intestinal Barrier Model by Modulation of Tight Junctions and Plasma Membrane Permeability"

_pharmaceutics, 2019, doi:10.3390/pharmaceutics11020073_

Round 1

Reviewer 1 Report

In this article, authors found new functions of the PN159 peptide, known as a cell-penetrating peptide. Authors found that the PN159 peptide modulates tight junctions of Caco-2 intestinal epithelial monolayer and enhances drug penetration. In addition, authors also determined antimicrobial activity of the PN159 peptide against wide range of bacteria. Data in this study are high levels in terms of quantity. This study will give beneficial information the readers and expand the research of drug delivery.

However, the following points should be considered to overcome several disputes regarding the outcome of the experimental data and author’s claims. If the following points are improved, the readers will be more benefited from this interesting article.

1) The most crucial point is that authors merely described about results in “Discussion” section. For example, in “Discussion 4.3”, authors didn’t discuss the relationship between modulation of tight junctions by the NP159 peptide and other experimental data (cellular uptake and antimicrobial activity). I didn’t understand the reason why authors determined the cellular uptake and antimicrobial activity of NP159 peptide in this study. Authors should explain about the relationship.

In “Discussion 4.4”, authors also had no discussion about why the modulation of tight junctions is NP159 peptide-specific function unlike other CPPs tested in this study. Authors should discuss the difference between the NP159 peptide and other CPPs.

2) In “Discussion 4.2”, authors explained the claudin-binding capacity as a common function in the NP159, C-CPE and C1C2 peptides, simply because of both peptides contain alpha-helix and beta-sheet structure. However, there are many peptides containing both alpha-helix and beta-sheet as common secondary structure. Authors should explain more information about structure similarity, for example peptide sequence similarity, in the NP159, C-CPE and C1C2 peptides.

Additionally, authors claim the NP159 peptide interacts with the claudin-4 and claudin-7 on the basis of docking studies. The data of docking studies strongly support author’s hypothesis. However, further studies indicating the direct interaction of NP159 peptide with claudin-4 and claudin-7, for example co-immunoprecipitation or co-localization analysis, are required to verify the author’s hypothesis.

3) In Figure 4, authors should explain about what arrow heads indicate.

4) In Figure 7A, authors should show not only merged image but also each fluorescent image because it is too difficult to recognize red fluorescence indicating ethidium-homodimer-1. Arrowheads indicating ethidium-homodimer-1 are also required to help readers to understand the membrane permeability by NP159 peptide. Additionally, the fluorescent images showing control cells are required in figure 7A and B.

Other careless mistakes were found in various points described below.

5) In line 52, authors should correct semicolons in reference numbers to commas.

6) In line 77, authors should delete a space between reference numbers.

7) In line85, authors should delete comma in rear of “the PN159 peptide”, and correct semicolon in rear of reference numbers to comma.

8) In line 95, authors should correct “E. coli” to “Escherichia coli” in italic letter.

9) In line 298, authors should delete “(Figure 1A)”.

10) In line 587, authors should correct “BBB” to “blood-brain barrier” because there is no explanation about the abbreviation.

11) In line 644, authors should correct “K. pneumoniae” to “Klebsiella pneumoniae”.

12) In line 709-710 about reference #8, authors should insert appropriate space in paper title.

Author Response

Point-by-point response to the reviews

First, we would like to thank the reviewers for their careful and constructive remarks. We are grateful for the opportunity to improve our manuscript. All the comments raised were addressed. The amendments are highlighted using yellow background in the revised version of the manuscript.

REVIEWER 1

1) The most crucial point is that authors merely described about results in “Discussion” section. For example, in “Discussion 4.3”, authors didn’t discuss the relationship between modulation of tight junctions by the PN159 peptide and other experimental data (cellular uptake and antimicrobial activity). I didn’t understand the reason why authors determined the cellular uptake and antimicrobial activity of PN159 peptide in this study. Authors should explain about the relationship.

Originally, we became interested in PN159 peptide when we searched for TJ modulators (Deli, 2009, Ref 1) and in our previous study we confirmed its permeability enhancing action on another barrier model and demonstrated its mode of action for the first time (Bocsik et al., 2016, Ref 8). We only later identified, with surprise, that the same peptide is known in other names, KLAL and MAP, in the literature as a model amphipathic CPP (Dathe et al., 1996, Ref 10). We wished to study both the TJ modulator and the membrane effects in the same study. However, there are no data whether these very different effects are connected, and if yes, how. To prove the connection a whole new set of experiments would be needed.

We added to Discussion 4.3.:

PN159, as a model amphipathic CPP peptide, strongly interacts with biological membranes due to its physical-chemical properties. As a CPP, PN159 enters epithelial cells (Mueller et al., 2008, Ref 42), enhances the membrane penetration of molecules (Oehlke et al., 1998, Ref 11) and exerts antimicrobial effects (Dathe, et al., 1996, Ref 10). Because PN159 binds different claudins and interacts specifically with their conserved amino acids, it acts as a TJ modulator (Bocsik et al., 2016, Ref 8). It is an open question, however, if the plasma membrane and the TJ modulator actions of PN159 are connected or not.

In “Discussion 4.4”, authors also had no discussion about why the modulation of tight junctions is PN159 peptide-specific function unlike other CPPs tested in this study. Authors should discuss the difference between the PN159 peptide and other CPPs.

As we demonstrated by our modeling results, PN159 peptide interacts with conserved residues of the investigated claudins (Figure 6) suggesting its ability for specific interactions (Bocsik et al., 2016, Ref 8). The amino acid sequence of PN159 results in an amphipathic helix structure (Dathe et al., 1996, Ref 10), which is not characteristic to the other CPPs investigated in our paper. From the other three investigated peptides, R8 and φ-peptide have positively charged arginine clusters (Table 1) and are considered as clearly cationic CPPs (Guidotti et al., 2017, Ref 17). Pep-1 also contains a positively charged amino acid cluster and is rich in tryptophan repeats (Table 1). PN159 neither contains clusters of arginine or positively charged amino acids, nor tryptophan (Table 1).

2) In “Discussion 4.2”, authors explained the claudin-binding capacity as a common function in the PN159, C-CPE and C1C2 peptides, simply because of both peptides contain alpha-helix and beta-sheet structure. However, there are many peptides containing both alpha-helix and beta-sheet as common secondary structure. Authors should explain more information about structure similarity, for example peptide sequence similarity, in the PN159, C-CPE and C1C2 peptides.

C-CPE peptide is the C-terminal fragment of the Clostridium perfringens enterotoxin, which preferentially binds claudin-3 and -4, thereby opens the paracellular cleft between intestinal epithelial cells (Deli, 2009, Ref 1). The synthetic peptide C1C2 is a novel peptidomimetic of the first extracellular loop of claudin-1, which specifically acts on claudin-1 and claudin-5, but not on other claudins (Dabrowski et al, 2015, Ref 40). The synthethic PN159 peptide binds claudin-1 and -5, but not claudin-3 (Bocsik et al., 2016, Ref 8), and in addition, interacts with claudin-4 and -7 based on our docking results in the present study.

We have added toDiscussion 4.2”:

What is common in the action of these TJ modulator peptides is that they have specific interactions with conserved amino acids of claudins (Dabrowski et al, 2015, Ref 40, Bocsik et al., 2016, Ref 8), however, these peptides do not share any similarity in their amino acid sequence.

Additionally, authors claim the PN159 peptide interacts with the claudin-4 and claudin-7 on the basis of docking studies. The data of docking studies strongly support author’s hypothesis. However, further studies indicating the direct interaction of PN159 peptide with claudin-4 and claudin-7, for example co-immunoprecipitation or co-localization analysis, are required to verify the author’s hypothesis.

We performed a co-localization analysis for claudin-4 and PN159 peptide. The “material and methods” was supplemented with the details of this new experiment.

We confirmed the interaction of PN159 peptide with claudin-4 junctional protein by a co-localization analysis. The labeled peptide was visible in the cytoplasm and also in the cell membrane at the level of intercellular junctions immunolabeled for claudin-4 (Figure 8).

Figure 8. Caco-2 cells were treated with Bodipy FL labeled PN159 peptide (5 min), then fluorescent immunostaining of junctional protein claudin-4 and staining of cell nuclei were performed. Red color: immunostaining for claudin-4. Green color: Bodipy FL labeled PN159 peptide. Blue color: staining of cell nuclei by H33342. Bar = 10 µm. Arrow heads indicate co-localization of claudin-4 and the labeled PN159 peptide.

3) In Figure 4, authors should explain about what arrow heads indicate.

Thank you for the remark, we completed the figure legend. Arrowheads indicate the opened junctions between the cells.

4) In Figure 7A, authors should show not only merged image but also each fluorescent image because it is too difficult to recognize red fluorescence indicating ethidium-homodimer-1. Arrowheads indicating ethidium-homodimer-1 are also required to help readers to understand the membrane permeability by PN159 peptide. Additionally, the fluorescent images showing control cells are required in figure 7A and B.

We completed Figure 7, as requested: we added control groups and showed separately the ethidium homodimer-1 images.

Figure 7. (A) Fluorescent staining of juntional protein zonula occludens-1 and double staining of cell nuclei in Caco-2 cells after 1-hour treatment with PN159 peptide. Green color: immunostaining for ZO-1. Red color: nucleus staining with ethidium homodimer-1 (EH-1), indicating increased membrane permeability. Blue color: staining of all cell nuclei by H33342. Bar = 10 µm. (B) Confocal microscopy images of living Caco-2 epithelial cells incubated with Bodipy FL labeled PN159 peptide in different concentrations for 5 min at 37°C. Green color: Bodipy FL-PN159 peptide. Blue color: staining of cell nuclei. Bar = 10 µm.

Other careless mistakes were found in various points described below.

5) In line 52, authors should correct semicolons in reference numbers to commas.

6) In line 77, authors should delete a space between reference numbers.

7) In line85, authors should delete comma in rear of “the PN159 peptide”, and correct semicolon in rear of reference numbers to comma.

8) In line 95, authors should correct “E. coli” to “Escherichia coli” in italic letter.

9) In line 298, authors should delete “(Figure 1A)”.

10) In line 587, authors should correct “BBB” to “blood-brain barrier” because there is no explanation about the abbreviation.

11) In line 644, authors should correct “K. pneumoniae” to “Klebsiellapneumoniae”.

12) In line 709-710 about reference #8, authors should insert appropriate space in paper title.

Thank you for the thorough reading of our manuscript, we corrected the mistakes.

Reviewer 2 Report

The submitted manuscript “Dual action of the PN159/KLAL/MAP peptide: increase of drug penetration across Caco-2 intestinal barrier model by modulation of tight junctions and plasma membrane permeability” by Bocsik et al have demonstrated the amphiphilic PN159 peptide can enhance its penetration on intestinal epithelial cells as well as TJ modulator. With a detailed study, the authors assessed the effect of PN159 peptide on drug intestinal epithelial cell viability, barrier function, its penetration activity and identified the potential targets via molecular modelling. They further determined the PN159 uptake activity and antimicrobial activity against nosocomial pathogens. Overall, this study is very well structured and interesting, I recommend it to be published on pharmaceutics after minor revision.

The authors should address the following comments before publishing.

1.    On page 11 line 420, it stated “they were elevated by 35-60 fold……”. But according to table 2, the calculated folds of FD (dextran 4-40KDa) are 159 (=3.5/0.022) to 400 (=6.0/0.015) folds. Similarly, it happened on page 19 line 595.

2.    On page 12 fig 4, there were several arrows in the picture. But there was a lack of definition in the caption. They should rectify it.

3.    On page 13 fig 5 B, the authors tested the CD structure at various temperature. They should label the numbers clearly with Celsius in Figure 5B.

4.    On page 16 line 525, they stated “the peptide was still effective on the vancomycin-resistant Enterobacter cloacae…….” When they used the ESKAPE pathogens, they should clearly state the bacteria strains including ATCC number in table 4. Additionally, they should also have a conventional antibiotic as the positive control for the antimicrobial test.

5.    On page 20 line 641, they mentioned another amphipathic antimicrobial peptide, Melittin, were used for the antimicrobial test. However, there is no such information in the main text. They should add their tested antimicrobial activity of melittin.

Author Response

Point-by-point response to the reviews

First, we would like to thank the reviewers for their careful and constructive remarks. We are grateful for the opportunity to improve our manuscript. All the comments raised were addressed. The amendments are highlighted using yellow background in the revised version of the manuscript.

REVIEWER 2

1. On page 11 line 420, it stated “they were elevated by 35-60 fold……”. But according to table 2, the calculated folds of FD (dextran 4-40KDa) are 159 (=3.5/0.022) to 400 (=6.0/0.015) folds. Similarly, it happened on page 19 line 595.

We agree, we also recalculated the fold changes and modified both the text and Table 2.

2. On page 12 fig 4, there were several arrows in the picture. But there was a lack of definition in the caption. They should rectify it.

Thank you for the remark, we completed the figure legend. Arrowheads indicate the opened junctions between the cells.

3. On page 13 fig 5 B, the authors tested the CD structure at various temperature. They should label the numbers clearly with Celsius in Figure 5B.

Thank you for noticing this, we completed Figure 5 as suggested.

4. On page 16 line 525, they stated “the peptide was still effective on the vancomycin-resistant Enterobacter cloacae…….” When they used the ESKAPE pathogens, they should clearly state the bacteria strains including ATCC number in table 4. Additionally, they should also have a conventional antibiotic as the positive control for the antimicrobial test.

We completed Table 4 with the ATCC number of the bacteria strains. Furthermore, a new experiment was performed with three antibiotics, cefoxitin, gentamicin and ciprofloxacin.

As compared to the reference bactericidal antibiotic cefoxitin, PN159 was more effective in the case of Acitenobacter baumannii, Enterococcus faecium, Enterobacter cloacae and Pseudomonas aeruginosa. Gentamicin and ciprofloxacin showed strong antimicrobial activity for all ESKAPE pathogens, except Enterococcus faecium. This strain was resistant for all investigated antibiotics, however, PN159 efficiently inhibited its growth (Table 4).

Table 4. Antibacterial activity of PN159 peptide, cefoxitin, gentamicin and ciprofloxacin antibiotics on ESKAPE pathogens. ATCC numbers are added in brackets. MIC (µM), minimal inhibitory concentration.

ESKAPE pathogens (ATCC)

PN159

Cefoxitin

Gentamicin

Ciprofloxacin

Acitenobacterbaumannii   (17978)

3.6

222.6

0.6

0.94

Enterococcus   faecium (700221)

4.4

>222.6

>143.9

>30.18

Staphylococcus   aureus (29213)

9.2

6.9

1.1

1.89

Klebsiella   pneumonia (10031)

13.8

3.5

0.6

0.06

Enterobacter   cloacae (13047)

31.0

>222.6

1.1

0.06

Pseudomonas   aeruginosa (27853)

46.4

>222.6

2.3

1.89

5. On page 20 line 641, they mentioned another amphipathic antimicrobial peptide, Melittin, were used for the antimicrobial test. However, there is no such information in the main text. They should add their tested antimicrobial activity of melittin.

We are sorry, if our text could be misunderstood. We have not tested melittin in our experiments. We wanted to discuss our antimicrobial results and compare them with data from the literature. Melittin was an interesting peptide for us, because it is an amphipathic peptide, like PN159, and in addition, a study (Maher et al., 2007, Ref 46) also indicated a TJ modulator effect of it.

Round 2

Reviewer 1 Report

Authors carefully responded to some issues pointed out. Authors are highly appreciated for the responses they made which are important for the readers as well as for the article itself. Revised points enhance author’s claim and this revised article is acceptable.